# Restricted and non-essential redundancy of RNAi and piRNA pathways in mouse oocytes

**Eliska Taborska**[1], **Josef Pasulka**[1], **Radek Malik**[1], **Filip Horvat**[1,2], **Irena Jenickova**[3], **Zoe Jelić Matošević**[2], **Petr Svoboda**[1]*

**1** Institute of Molecular Genetics of the Czech Academy of Sciences, Prague 4, Czech Republic,
**2** Bioinformatics Group, Division of Molecular Biology, Department of Biology, Faculty of Science, University of Zagreb, Zagreb, Croatia, **3** Czech Centre of Phenogenomics, Institute of Molecular Genetics of the Czech Academy of Sciences, Vestec, Czech Republic

* svobodap@img.cas.cz

**Data Availability Statement:** RNA-seq data were deposited in the Gene Expression Omnibus database under accession ID GSE132121.

## Abstract

Germline genome defense evolves to recognize and suppress retrotransposons. One of defensive mechanisms is the PIWI-associated RNA (piRNA) pathway, which employs small RNAs for sequence-specific repression. The loss of the piRNA pathway in mice causes male sterility while females remain fertile. Unlike spermatogenic cells, mouse oocytes posses also RNA interference (RNAi), another small RNA pathway capable of retrotransposon suppression. To examine whether RNAi compensates the loss of the piRNA pathway, we produced a new RNAi pathway mutant *Dicer*^SOM and crossed it with a catalytically-dead mutant of *Mili*, an essential piRNA gene. Normal follicular and oocyte development in double mutants showed that RNAi does not suppress a strong ovarian piRNA knock-out phenotype. However, we observed redundant and non-redundant targeting of specific retrotransposon families illustrating stochasticity of recognition and targeting of invading retrotransposons. Intracisternal A Particle retrotransposon was mainly targeted by the piRNA pathway, MaLR and RLTR10 retrotransposons were targeted mainly by RNAi. Double mutants showed accumulations of LINE-1 retrotransposon transcripts. However, we did not find strong evidence for transcriptional activation and mobilization of retrotransposition competent LINE-1 elements suggesting that while both defense pathways are simultaneously expendable for ovarian oocyte development, yet another transcriptional silencing mechanism prevents mobilization of LINE-1 elements.

## Author summary

Retrotransposons are mobile genomic parasites causing mutations. Germ cells need protection against retrotransposons to prevent heritable transmission of their new insertions. The piRNA pathway is an ancient germline defense system analogous to acquired immunity: once a retrotransposon jumps into a piRNA-producing locus, which provides a kind of a "genomic sensor" for actively transposing elements, it is recognized and suppressed. Remarkably, the murine piRNA pathway is essential for spermatogenesis but not oocyte development. In contrast, zebrafish lacking the piRNA pathway do not develop any germ

**Funding:** This work was funded from the European Research Council under the European Union's Horizon 2020 research and innovation programme (grant agreement No 647403, D-FENS). Additional support was provided by the Ministry of Education, Youth, and Sports (MEYS) project NPU1 LO1419 (Biomodels for health). Financial support of E.T. and F.H. was in part provided by the Charles University through a PhD student fellowship; this work will be in part used to fulfill requirements for a PhD degree and hence can be considered "school work". FH and ZJM were supported by the European Structural and Investment Funds grant for the Croatian National Centre of Research Excellence in Personalized Healthcare (contract #KK.01.1.1.01.0010), Croatian National Centre of Research Excellence for Data Science and Advanced Cooperative Systems (contract #KK.01.1.1.01.0009) and Croatian Science Foundation (grant IP-2014-09-6400). Production and histology analysis of the DicerSOM mice was supported by RVO 68378050 by Academy of Sciences of the Czech Republic and by LM2015040 (Czech Centre for Phenogenomics), CZ.1.05/ 2.1.00/19.0395 (Higher quality and capacity for transgenic models), CZ.1.05/1.1.00/02.0109 (BIOCEV - Biotechnology and Biomedicine Centre of the Academy of Sciences and Charles University). Microscopy was done at the Light Microscopy Core Facility, IMG CAS, Prague, Czech Republic supported by LM2015062 (Czech-Bioimaging). Additional computational resources were provided by CESNET LM2015042. The funders had no role in study design, data collection and analysis, decision to publish, or preparation of the manuscript.

**Competing interests:** The authors have declared that no competing interests exist.

cells. It was hypothesized that RNA interference pathway could rescue oocyte development in mice lacking the piRNA pathway. RNA interference also targets retrotransposons and is particularly enhanced in mouse oocytes. To test this hypothesis, we engineered mice lacking both pathways and observed that oocytes in these mice develop normally, which argues against the hypothesis. Furthermore, analysis of individual retrotransposon groups revealed that in specific cases the two pathways mutually compensate each other. However, this redundancy apparently evolved stochastically and is restricted to specific retrotransposon groups. Finally, our results indicate that there must be yet another layer of retrotransposon silencing in mouse oocytes, which prevents high retrotransposon activity in the absence of piRNA and RNA interference pathways.

## Introduction

Genome integrity in the germline is important for intact transmission of genetic information into progeny. However, it is being disturbed by retrotransposons, mobile genomic parasites reproducing through a "copy & paste" strategy (reviewed in [1]). Although retrotransposons threaten genome integrity, they occasionally also provide new functional gene elements, such as promoters, enhancers, exons, splice junctions, or polyA signals, thus contributing to evolution of new traits (reviewed in [1]). Retrotransposons can be categorized by the presence of long terminal repeats (LTRs) and retrotransposition autonomy [2]. While contributions of different families vary, their collective contribution to the mammalian genome content is substantial: annotatable retrotransposon insertions comprise about a half of human and mouse genomes [3, 4]. The mouse genome hosts two mobile autonomous retrotransposons: non-LTR element LINE-1 (L1) and LTR element Intracisternal A Particle (IAP) [5–7]. L1 (reviewed in [8]) is the most successful mammalian autonomous retrotransposon with 868,000 insertions in the mouse genome [4]. L1 has several unique adaptations, such as a unique complex bidirectional promoter [9, 10] and retrotransposition in *cis* [11]. IAP is an endogenous retrovirus, which invaded the mouse genome relatively recently [6]. RepeatMasker identifies approximately 13,000 IAP insertions, of which 44% represent solo LTRs [12].

Various transcriptional and post-transcriptional mechanisms evolved to repress retrotransposons (reviewed in [13]). A key mechanism suppressing retrotransposons in the mammalian germline is the PIWI-interacting RNA (piRNA) pathway, which combines post-transcriptional and transcriptional silencing (reviewed in [14]). The piRNA pathway relies on specific genomic regions (piRNA clusters) that are sensing invading mobile elements and giving rise to piRNAs. These are 25–30 nucleotides long RNAs loaded onto PIWI subgroup of the Argonaute protein family, which guide retrotransposon recognition and repression [15]. A complex piRNA system starts with processing primary transcripts from piRNA clusters into piRNAs, which guide cleavage of retrotransposon transcripts. This triggers production of a secondary piRNA pool, which further facilitate processing of primary transcripts into piRNAs (reviewed in [14]).

Mice use three PIWI proteins PIWIL1 (MIWI), PIWIL2 (MILI), and PIWIL4 (MIWI2). PIWIL3, the fourth mammalian PIWI protein, was found in bovine oocytes [16] but *Piwil3* gene was apparently lost in the common ancestor of mice and rats. All three mouse PIWI proteins are essential for spermatogenesis but not oogenesis [17–19] although the piRNA pathway operates during oogenesis where it targets retrotransposon expression [20, 21]. MILI is a cytoplasmic protein, which generates primary piRNAs and secondary piRNAs by so-called ping-pong mechanism with itself or with MIWI2 [22–24]. MIWI2 shuttles to the nucleus and

mediates transcriptional silencing through DNA and histone methylation of retrotransposon loci [25, 26]. In males, MIWI2 expression ceases around birth whereas MILI is important for clearance of retrotransposon transcripts in postnatal testes [24, 27]. MILI and MIWI2 cooperate on silencing of L1 and IAP (non-LTR and LTR) retrotransposons in mouse fetal testes [28, 29].

It is unknown what accounts for the strikingly different phenotypes of piRNA pathway mutants in murine male and female germlines. In contrast to mice, *Drosophila* or zebrafish females lacking piRNA pathway components are sterile [17–19, 30]. It was hypothesized that the loss of the piRNA pathway in the mouse female germline could be compensated by RNA interference (RNAi) [31, 32]. Canonical RNAi is defined as sequence-specific RNA degradation induced by long double-stranded RNA (dsRNA) [33] and is active in mouse oocytes [34, 35]. The canonical RNAi starts with processing of long dsRNA by RNase III Dicer into ~22 nucleotides long small interfering RNAs (siRNAs), which guide sequence-specific cleavage of perfectly complementary RNAs by Argonaute 2 (AGO2) (reviewed in [36]). Endogenous RNAi in mouse oocytes and zygotes was shown to target mobile elements and regulate gene expression [31, 32, 37–40]. For example, L1 can be targeted in oocytes by piRNAs and endo-siRNAs [32, 40–43]. Analysis of small RNA sequencing (RNA-seq) data from mouse oocytes [43] shows that piRNA and RNAi pathways may target retrotransposons simultaneously but the extent of repression for individual retrotransposons by each pathway could vary (Fig 1). Importantly, endogenous RNAi in mouse oocytes evolved during rodent evolution through exaptation of an MT retrotransposon insertion in intron 6 of *Dicer* gene, which functions as an oocyte-specific promoter producing a unique truncated Dicer isoform (denoted Dicer$^O$). *Dicer$^O$* transcripts accumulate during oocyte's growth and persist into the zygote [44]. Based on RNA-seq, *Dicer$^O$* is expressed already in non-growing oocytes 5 days postpartum (5dpp) [45].

To address the significance of RNAi and piRNA pathway redundancy, we produced and analyzed mice lacking both pathways. We first produced a novel mouse model lacking *Dicer$^O$* (denoted *Dicer$^{SOM}$*) and then crossed it with *Mili$^{DAH}$* mice expressing catalytically inactive MILI [24]. We show that engagement of piRNA and RNAi pathway in repression of retrotransposons is stochastic and occasionally synergic. The simultaneous loss of RNAi and piRNA pathways uncovers redundant targeting of L1 and extended L1 derepression but does not affect oocyte development. Thus, while piRNAs are essential for spermatogenesis and show restricted redundancy with RNAi, they do not have essential role in ovarian oocyte development that would be masked by RNAi.

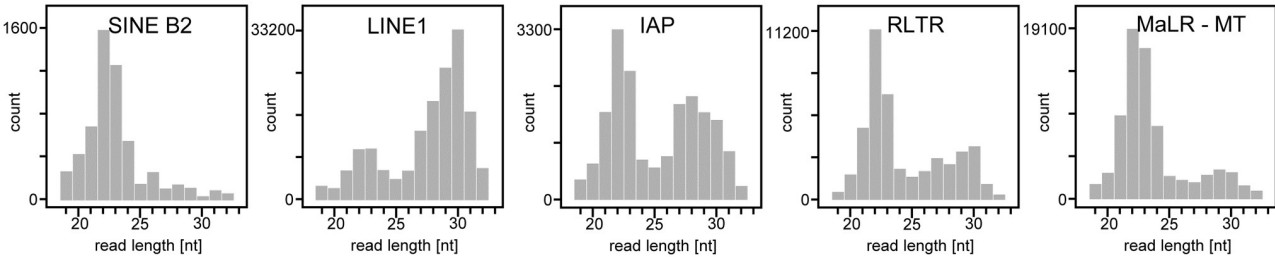

**Fig 1. Retrotransposon-derived small RNAs.** Different retrotransposons are associated with different length distribution of small RNAs from mouse oocytes suggesting partially redundant repression of retrotransposons by endogenous RNAi and piRNA pathways. Based on RNA-seq data from Yang et al., 2016 [43].

## Results

### Generation of Dicer^SOM mice—A novel RNAi-deficient model

The oocyte-specific $Dicer^O$ isoform comprises the majority of Dicer protein in mouse oocytes and deletion of its LTR-derived promoter ($Dicer^{\Delta MT}$ mutant) phenocopies conditional $Dicer$ knock-out in mouse oocytes [38, 44]. However, we have discovered that there is a second LTR insertion (MTA) in intron 6 (Fig 2A), which can also produce a truncated Dicer variant and reduce phenotype penetrance in $Dicer^{\Delta MT/\Delta MT}$ mice [46]. Thus, to eliminate truncated Dicer expression in mouse oocytes, we generated another modified $Dicer$ allele (denoted $Dicer^{SOM}$), which has an HA-tag at the N-terminus and lacks introns 2–6 (Fig 2A). It was shown that N-terminal HA-tag on Dicer does not produce a phenotype and allows for its detection with highly-specific anti-HA antibodies [47].

$Dicer^{SOM}$ allele was generated in mouse embryonic stem cells (ESCs) by cleaving intron 2 and intron 6 with Cas9 nuclease followed by homologous recombination with a construct carrying an HA-tag fused to the 5' end of $Dicer$ cDNA containing coding sequence of exon 2 to

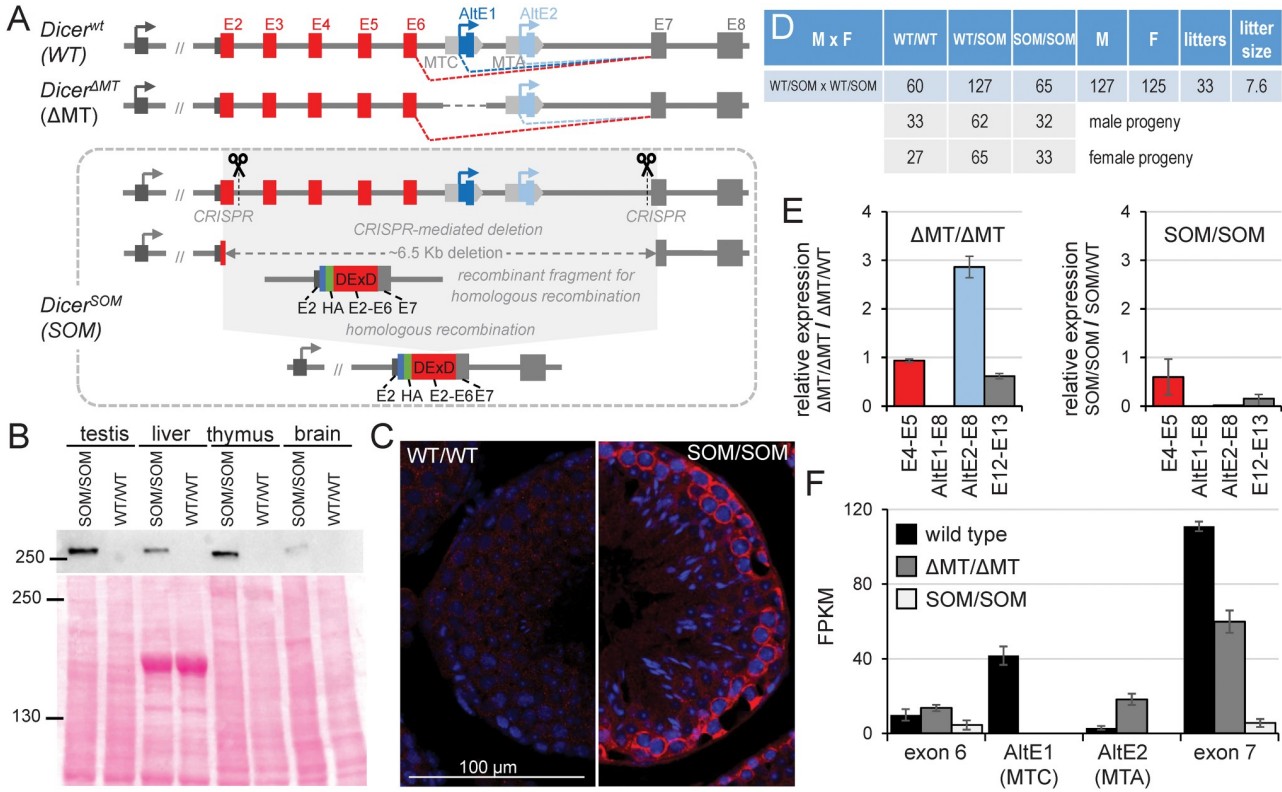

**Fig 2. $Dicer^{SOM}$ mouse model.** (A) Schematic depiction of the 5' gene structure of $Dicer$ and $Dicer^{Y\Delta MT}$ and $Dicer^{SOM}$ models. $Dicer^{SOM}$ model was produced by removing ~6.5 kb of genomic DNA using CRISPR nucleases followed by homologous recombination with a construct encoding an HA-tag and exons 2–7. The resulting allele lacks introns 2 to 6 while encoding HA-tagged full-length Dicer. (B) $Dicer^{SOM}$ mice express HA-tagged full-length Dicer. 40 μg of total protein lysate were loaded per lane and loading consistency was confirmed by Ponceau S staining of the Western blot membrane (lower panel). (C) Dicer^{SOM} protein has cytoplasmic expression. Shown are sections of seminiferous tubules stained with anti-HA antibody revealing cytoplasmic Dicer^{SOM} signal (red color) in spermatogonia of $Dicer^{SOM/SOM}$ mice. DNA was stained with DAPI (blue color). (D) $Dicer^{SOM/SOM}$ animals are born in a Mendelian ratio upon crossing $Dicer^{SOM/wt}$ parents. (E) Analysis of Dicer expression by qPCR in oocytes of homozygous $Dicer^{\Delta MT}$ and $Dicer^{SOM}$ mutants. Bar colors indicate cDNA region amplified by primers localized in exons as described in the x-axis legend. $Dicer^{\Delta MT/\Delta MT}$ mice show loss of expression of the MTC-driven Dicer variant and a relative increase of expression from the downstream MTA LTR insertion. $Dicer^{SOM/SOM}$ mice show complete loss of short Dicer isoform. Data from three biological replicates were normalized to oocytes from heterozygous littermates. Error bar = SD. (F) Analysis of Dicer expression in oocytes of wild-type animals and homozygous $Dicer^{\Delta MT}$ and $Dicer^{SOM}$ mutants by RNA-seq (three biological replicates). Depicted are RPKM values per exon. Error bar = SD.

exon 7. Homologous recombination arms contained intron 1 and intron 7 (Fig 2A). An ESC line carrying the desired recombination event (S1 Fig) was subsequently used to make mouse chimaeras from which we established a mouse line. Analysis of $Dicer^{SOM}$ mice showed that the full-length HA-tagged Dicer could be detected by Western blotting in organ lysates (Fig 2B) and by immunofluorescent staining of histological sections albeit the signal was relatively low with the protocol and antibody used (Fig 2C). Breeding of $Dicer^{SOM/wt}$ animals yielded progeny with $Dicer^{SOM/SOM}$ genotypes at the expected Mendelian ratio (Fig 2D). $Dicer^{SOM/SOM}$ mice appeared normal and males were fertile suggesting that the introduced modification did not perturb normal full-length Dicer function. Importantly, $Dicer^{SOM/SOM}$ females were sterile, which was the expected phenotype caused by the lack of $Dicer^O$. Analysis of $Dicer^{SOM/SOM}$ oocytes by qPCR and RNA-seq showed the loss of $Dicer^O$-encoding transcript and minimal levels of the full-length Dicer isoform mRNA, which contrasted with $Dicer^{\Delta MT/\Delta MT}$ oocytes where expression from the MTA LTR was enhanced (Fig 2E and 2F). Thus, $Dicer^{SOM/SOM}$ mice solved the problem with residual $Dicer^O$ expression observed in $Dicer^{\Delta MT/\Delta MT}$ mice.

## $Dicer^{SOM}$ mutants phenocopy oocyte-specific $Dicer$ knock-out

Next, we analyzed the phenotype of $Dicer^{SOM/SOM}$ oocytes and compared that with known phenotypes of $Dicer$ and $Ago2$ mutants, which all exhibit sterility and spindle defects while having different impact on oocyte's small RNA biogenesis and suppression of their targets [31, 38, 44]. Fully-grown $Dicer^{SOM/SOM}$ oocytes showed high frequency of meiotic spindle defects upon resumption of meiosis (Fig 3A and 3B and S2 Fig) and upregulated mRNA levels of RNAi targets (Fig 3C), consistent with RNAi pathway deficiency. For precise characterization of transcriptome changes in $Dicer^{SOM/SOM}$ oocytes, we performed RNA-seq of fully-grown germinal vesicle-intact (GV) oocytes. When comparing mRNA changes in $Dicer^{\Delta MT/\Delta MT}$ and $Dicer^{SOM/SOM}$ oocytes, the later showed more deregulated transcriptome (Fig 3D). In addition, we observed upregulated mRNAs of many genes, which would have potential to be targeted by RNAi (Fig 3D, red points), i.e. genes where complementary endo-siRNAs were identified in RNA-seq datasets from mouse oocytes [32, 41, 43]. This was consistent with the principal component analysis (PCA) where $Dicer^{\Delta MT/\Delta MT}$ samples localized between wild-type and $Dicer^{SOM/SOM}$ samples (Fig 3E). We included into PCA also transcriptomes of previously analyzed $Dicer$ and $Ago2$ mutants [31]. Although the experimental variability ("bench effect") separated our and other datasets along the PC1 axis, the data showed an apparent separation of wild-type and RNAi mutant samples along the PC2 axis. $Dicer^{SOM/SOM}$ transcriptome appeared to be more distant from wild-type samples than $Dicer^{\Delta MT/\Delta MT}$ transcriptome and in a parallel position with Stein et al. mutants [31] relative to wild-type samples (Fig 3E).

When comparing relative transcriptome changes of Dicer mutants to accompanying wild-type controls (Fig 3F and S3 Fig), $Dicer^{\Delta MT/\Delta MT}$ and $Dicer^{SOM/SOM}$ showed good overall correlation with transcriptome changes in $Dicer^{-/-}$ oocytes (r = 0.75) and relative changes of predicted RNAi targets correlated slightly better in $Dicer^{SOM/SOM}$ (r = 0.89) than changes in $Dicer^{\Delta MT/\Delta MT}$ (r = 0.86) oocytes. While the difference in correlation coefficient is minimal, putative RNAi targets and other upregulated genes tend to be distributed above the diagonal when comparing $Dicer^{\Delta MT/\Delta MT}$ and $Dicer^{-/-}$ samples and more along it when comparing $Dicer^{SOM/SOM}$ with $Dicer^{-/-}$ (Fig 3F). This means that transcriptome changes in $Dicer^{SOM/SOM}$ better mimic changes in $Dicer^{-/-}$ than transcriptome changes in $Dicer^{\Delta MT/\Delta MT}$ and show that $Dicer^{SOM/SOM}$ model is superior to $Dicer^{\Delta MT/\Delta MT}$ as oocyte-specific RNAi deficiency model. As RNAi also targets repetitive sequences, we analyzed correlations of relative changes of retrotransposon RNAs in our datasets and $Dicer$ and $Ago2$ mutants [31]. In this case, $Dicer^{SOM/SOM}$ mutants correlated with $Dicer^{-/-}$ mutant (r = 0.90) even better than $Dicer^{-/-}$

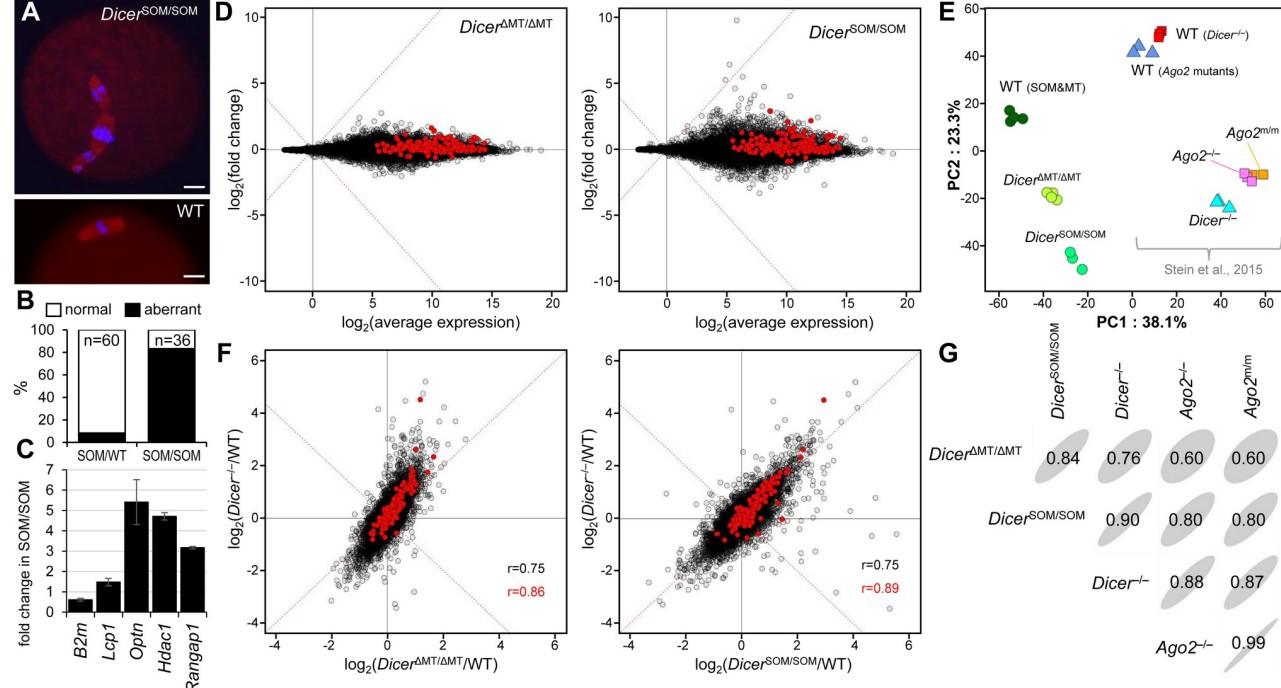

**Fig 3. *Dicer*<sup>SOM/SOM</sup> animals phenocopy oocyte-specific Dicer knock-out.** (A) Oocytes of *Dicer*<sup>SOM/SOM</sup> mice exhibit spindle defects like *Dicer*<sup>−/−</sup> or *Dicer*<sup>ΔMT/ΔMT</sup> oocytes [38, 39, 44]. Size bar = 10 μm. (B) Frequency of meiotic spindle defects in oocytes from *Dicer*<sup>SOM/wt</sup> and *Dicer*<sup>SOM/SOM</sup> mice. Numbers correspond to examined oocytes with each genotype. (C) mRNA expression of selected RNAi targets analyzed by qPCR performed as three biological replicates. *B2m* is a non-targeted gene. All mRNA levels are shown relative to *Hprt1*. (D) MA plots depicting transcriptome changes in *Dicer*<sup>ΔMT/ΔMT</sup> and *Dicer*<sup>SOM/SOM</sup> oocytes relative to wild-type oocytes. (E) PCA analysis of RPKM transcriptome changes in oocytes of different mutants, which includes previously published data from *Dicer* and *Ago* mutants [31]. (F) Comparison of mRNA expression changes in *Dicer*<sup>ΔMT/ΔMT</sup> and *Dicer*<sup>SOM/SOM</sup> oocytes (x-axis) with mRNA changes in oocytes with conditional Dicer knock-out (y-axis). (G) Correlation matrices of retrotransposon-derived transcriptome in different mutant oocytes. Elliptic shapes reflect sizes of inscribed correlation coefficients for easier visual navigation.

mutant with *Ago2* mutants (Fig 3G), which is remarkable considering entirely independent experimental analysis. Since *Dicer*<sup>SOM/SOM</sup> females closely phenocopy conditional *Dicer* knock-out in oocytes, the *Dicer*<sup>SOM</sup> model is excellent for studying simultaneous loss of RNAi and piRNA pathways in oocytes because it overcomes arduous breeding of the *Mili*<sup>DAH</sup> model with a conditional *Dicer* knock-out.

## RNAi & piRNA mutant females show no signs of disturbed ovarian development of oocytes

Mouse mutants lacking piRNA and RNAi pathways in mouse oocytes should reveal whether or not mutants of single pathways manifest the full extent of small RNA-mediated retrotransposon repression because both pathways can suppress mobile elements. Accordingly, we crossed *Dicer*<sup>SOM</sup> mice with *Mili*<sup>DAH</sup> mice carrying a mutation in the DDH catalytic triad of MILI where the second aspartic acid is mutated to an alanine [24]. As *Mili*<sup>DAH/DAH</sup> males and *Dicer*<sup>SOM/SOM</sup> females are sterile, we first crossed *Mili*<sup>DAH/DAH</sup> females to *Dicer*<sup>SOM/SOM</sup> males (both on C57Bl/6 background) and then crossed their double heterozygote progeny to obtain *Dicer*<sup>SOM/WT</sup>, *Mili*<sup>DAH/DAH</sup> females and Dicer<sup>SOM/SOM</sup>, *Mili*<sup>DAH/WT</sup> males, which were crossed to obtain females lacking in oocytes both pathways (*Dicer*<sup>SOM/SOM</sup>, *Mili*<sup>DAH/DAH</sup>, "double KO"), only RNAi (*Dicer*<sup>SOM/SOM</sup>, *Mili*<sup>DAH/WT</sup>), only piRNA (*Dicer*<sup>SOM/WT</sup>, *Mili*<sup>DAH/DAH</sup>), or none (*Dicer*<sup>SOM/WT</sup>, *Mili*<sup>DAH/WT</sup>). Importantly, phenotype analysis of double mutants was

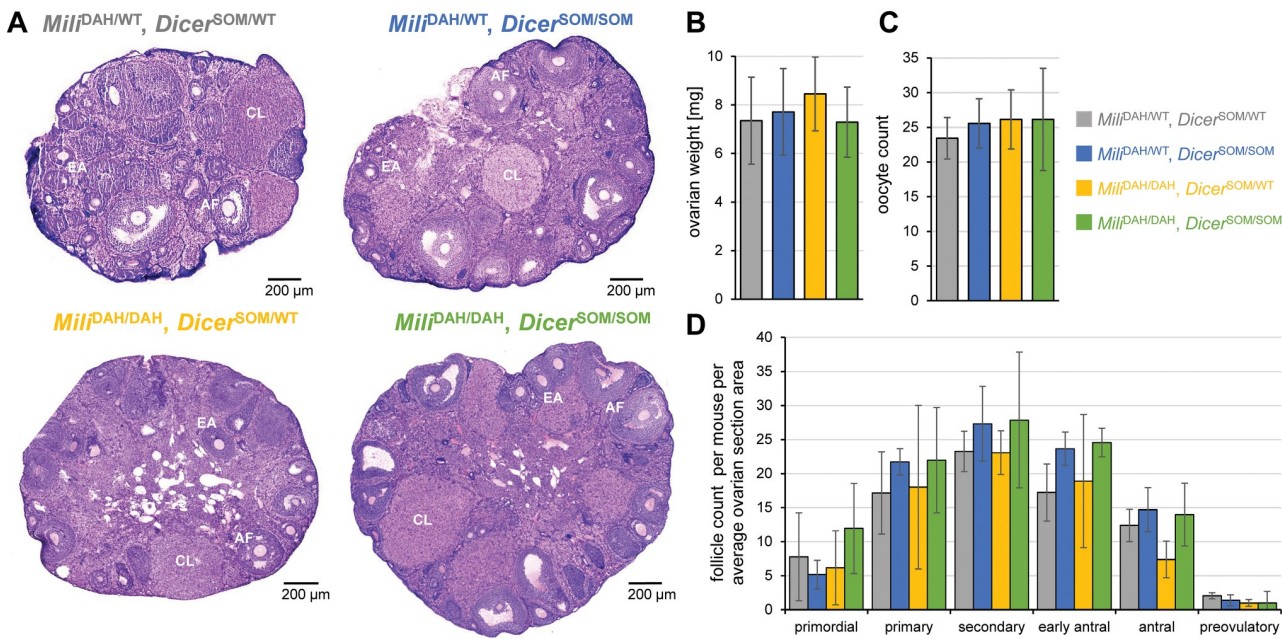

**Fig 4. RNAi (Dicer^SOM/SOM) and piRNA (Mili^DAH/DAH) knock-out analysis.** (A) Representative histological sections of ovaries stained with hematoxylin and eosin. Abbreviations: CL–*corpus luteum*, EA—early antral follicle, AF—antral follicle. (B) Ovarian weight. Ten ovaries from five 14–19 weeks-old females were analyzed per genotype. (C) Yield of fully-grown germinal vesicle (GV)-intact oocytes per animal. Seven 10–16 weeks old females per genotype were analyzed. (D) Analysis of follicles in ovaries of mutant animals. Follicular content has been normalized per average ovarian section area per animal. Six ovaries from three animals were analyzed. Error bar = SD.

restricted to ovarian phenotypes manifested until the fully-grown GV oocyte stage because the loss of RNAi causes meiotic spindle defects in ovulated oocytes and impairs further development [38, 39, 44].

Females of the four above-described genotypes were analyzed for ovarian weight and histology and number of fully-grown GV oocytes recovered from non-superovulated animals (Fig 4A–4C). Double KO females showed normal ovarian morphology and weight of ovaries (Fig 4A and 4B). The presence of antral follicles and *corpora lutea* in ovaries of double KO females suggested that the loss of RNAi on the piRNA knock-out background does not reveal a strong ovarian phenotype. This was further supported by recovering normal number of fully-grown oocytes from double KO ovaries (Fig 4C). Elimination of both pathways thus did not significantly affect follicular growth and ovarian oocyte development (Fig 4D), which shows that RNAi pathway does not mask an essential function of piRNAs during ovarian oogenesis.

## Varying redundancy of retrotransposon repression by RNAi and piRNA pathways

To complete the analysis of double mutants, we examined retrotransposon transcript levels. Previous studies showed upregulation of specific retrotransposon types in single pathway mutants but the extent to which retrotransposons are simultaneously regulated and how one pathway could compensate for the loss of the other one was not clear. We analyzed by qPCR active subfamilies of autonomous retrotransposon L1 (L1Md_T) and IAP (IAPEz) and selected non-autonomous elements SINE B1, RLTR10, and MTA. Comparison of oocytes of the four genotypes (double KO, RNAi KO, piRNA KO, and double heterozygote) showed interesting diversity of transcript level changes (Fig 5A).

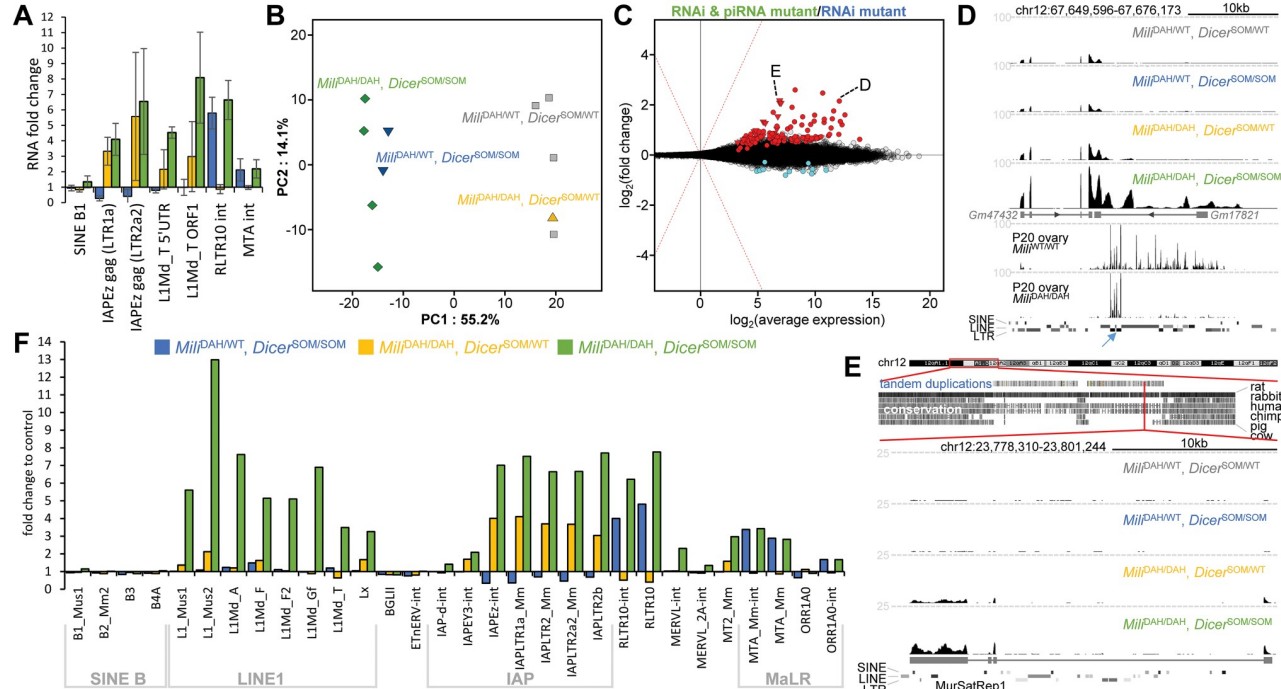

**Fig 5. Transcriptional changes in oocytes lacking RNAi and piRNA pathways.** (A) qPCR analysis of selected retrotransposons in mutant oocytes. Retrotransposon RNA levels are shown relative to that in heterozygous ($Mili^{\mathrm{DAH/WT}}$, $Dicer^{\mathrm{SOM/WT}}$) oocytes. Three biological replicates were perfomed. Error bar = SD. (B) PCA analysis of RPKM transcriptome changes in oocytes with mutated RNAi ($Mili^{\mathrm{DAH/WT}}$, $Dicer^{\mathrm{SOM/SOM}}$), piRNA ($Mili^{\mathrm{DAH/DAH}}$, $Dicer^{\mathrm{SOM/WT}}$) or both pathways ($Mili^{\mathrm{DAH/DAH}}$, $Dicer^{\mathrm{SOM/SOM}}$). Oocytes from heterozygous littermates ($Mili^{\mathrm{DAH/WT}}$, $Dicer^{\mathrm{SOM/WT}}$) served as controls. (C) MA plot depicting transcriptome changes in double mutant oocytes relative to RNAi pathway mutant oocytes. MA plots depicting transcriptome changes of pathway mutants relative to heterozygous control are provided in S3 Fig. Transcripts with significantly higher and lower transcript levels are shown in red and blue, respectively and are provided in the Supplementary file 1. Transcripts from a tandem duplication cluster at chromosome 12 are depicted as triangles. One such a transcript was chosen for display in the panel E. Transcript labeled with D is associated with a $Mili$-dependent small RNA cluster. (D) A UCSC browser snapshot of a region producing $Mili$-dependent and $Mili$-independent small RNAs. The locus encodes non-coding RNAs ($Gm476432$ and $Gm17821$), which show increased abundance in double mutants. The upper part shows RNA-seq data from oocytes with the four genotypes. Dashed horizontal lines depict expression level corresponding to 100 counts per million (CPM). The lower two samples show small RNAs from wild-type and $Mili$-mutant postnatal ovaries (day 20) from Kabayama et al. [21]. The $Mili^{\mathrm{DAH/DAH}}$ mutant is the same one as in our study. The bottom part depicts distribution of repetitive elements along the locus with a blue arrow pointing to the LTRIS_Mm inverted repeat giving rise to $Mili$-independent 21–23 nt small RNAs. (E) An example of one of the paralogs from the tandem duplication cluster at chromosome 12, which exhibit increased transcript levels in double mutants. The upper part shows the tandem duplication cluster at chromosome 12, the lower half shows a snapshot from a UCSC browser with scaled RNA-seq data from different mutants. Expression analysis data from UCSC browser are organized as in the panel D. The bottom part depicts distribution of SINE, LINE and LTR retrotransposons and MurSatRep1 repeat along the locus. (F) RNA abundance of selected retrotransposon types from Repeatmasker annotation in mutant oocytes are displayed relative to heterozygous ($Mili^{\mathrm{DAH/WT}}$, $Dicer^{\mathrm{SOM/WT}}$) controls.

IAP seemed to be targeted by the piRNA pathway and was even better suppressed in RNAi knock-out oocytes. Levels of IAPs internal sequence showed low if any increase when RNAi was lost in addition to the piRNA pathway (Fig 5A). A strong effect of double knock-out was observed on L1Md_T expression. While RNAi knock-out had no effect and $Mili$ knock-out showed only slightly increased L1 expression, the loss of both pathways caused about six-fold increase in expression suggesting that both pathways target L1 and can mutually compensate loss of a single pathway. In contrast, RNAi primarily targeted RLTR10. $Mili$ knock-out had no effect on RLTR10 expression on its own but had minor additive effect on $Dicer^{\mathrm{SOM/SOM}}$ background (Fig 5A). These data suggest that RNAi and piRNA pathways suppress some but not all retrotransposons in a redundant manner and can partially compensate for each other's loss.

To obtain an independent confirmation of qPCR data, we performed RNA-seq analysis of the four genotypes. The first component of a PCA analysis showed that transcriptomes of

double mutants were quite similar to RNAi mutants while the loss of that piRNA pathway had a minor effect (Fig 5B). These results were consistent with a more detailed survey of transcriptome changes in individual mutants (S4 Fig). Given the PCA results and our interest in understanding of the phenotype of the double mutant, we focused on transcriptome changes in double mutant oocytes relative to RNAi mutants. This analysis revealed that a small population of genes (127 genes) exhibited significantly increased transcript abundance on top of gene expression changes caused by the loss of RNAi (Fig 5C and S1 Table).

The most interesting locus exemplifying a gene coupling RNAi and piRNA pathways included lncRNAs *Gm47732* add *Gm17821* (Fig 5D). Both lncRNAs showed increased transcript abundance in double mutants. *Gm17821* apparently overlaps with a primary unidirectional piRNA cluster as evidenced by *Mili*-dependent maternal piRNAs [21]. At the same time, *Gm17821* intron contains an inverted repeat of two LTRIS_Mm sequences, which gives rise to *Mili*-independent 21–23 nt small RNAs (Fig 5D). Therefore, this locus produces primary piRNA transcripts, which are also processed by RNAi machinery.

Other genes, which had increased expression in double mutants were a mixture of protein coding genes and long non-coding RNAs. Although many transcripts carried L1 or IAP sequences in 3'UTRs, it is unclear whether they represent genes simultaneously regulated by RNAi and piRNA pathways. One interesting observation worth of further investigation was increased abundance of over twenty long non-coding RNA (lncRNA) distinct paralogs, which carried MurSatRep1 repetitive sequence in their last exon and localized into a large (6.7 MB) and complex segmental duplication locus on mouse chromosome 12 (Fig 5E).

Analysis of repetitive sequences in RNA-seq data also corroborated previous qPCR analysis (Fig 5F). This included dominating piRNA-dependent suppression of IAP elements where the loss of RNAi even improved suppression of different IAP subgroups while the loss of both pathways had caused higher IAP expression than the loss of the piRNA pathway alone. RLTR10 and MaLR elements were primarily targeted by RNAi while L1 elements showed the strongest redundancy of RNAi and piRNA pathways; only a combined suppression of both pathways yielded increased abundance of L1 sequence (Fig 5F). RNA-seq data thus confirmed that RNAi and piRNA pathways suppress some but not all retrotransposons in a redundant manner and compensate for each other's loss particularly in the case of L1 elements. At the same time, even when both pathways simultaneously fail to suppress retrotransposons in non-growing oocytes or during oocyte growth, this does not lead to a failure of ovarian oogenesis.

## Minimal expression of autonomous retrotransposition-competent L1s in mouse oocytes

The lack of sterile phenotype in female mice upon removal of two pathways targeting retrotransposons could stem from specific behavior/repression of DNA-damaging retrotransposons in the mouse female germline, which would make it less sensitive to the loss of the piRNA pathway. Expression of retrotransposons can differ between male and female germlines as retrotransposons adapt their transcriptional control to transcription factors regulating gene expression during different parts of the germline cycle. It can be observed in RNA-seq data [45, 48–50]. Mouse oocytes have relatively low expression of L1 and IAP (Fig 6A). Mouse genome contains several L1 families, of which LINE1_Md_A, LINE1_Md_T(-F) and LINE1_Md_G(-F) are the most recent and contain full-length retrotransposition-competent LINE1 copies [51–53]. Remarkably, despite the number of genomic insertions, the contribution of L1 sequences from these subfamilies to the oocyte transcriptome is relatively low, generally not exceeding 50 reads per million (Fig 6B). While the double mutants show about an

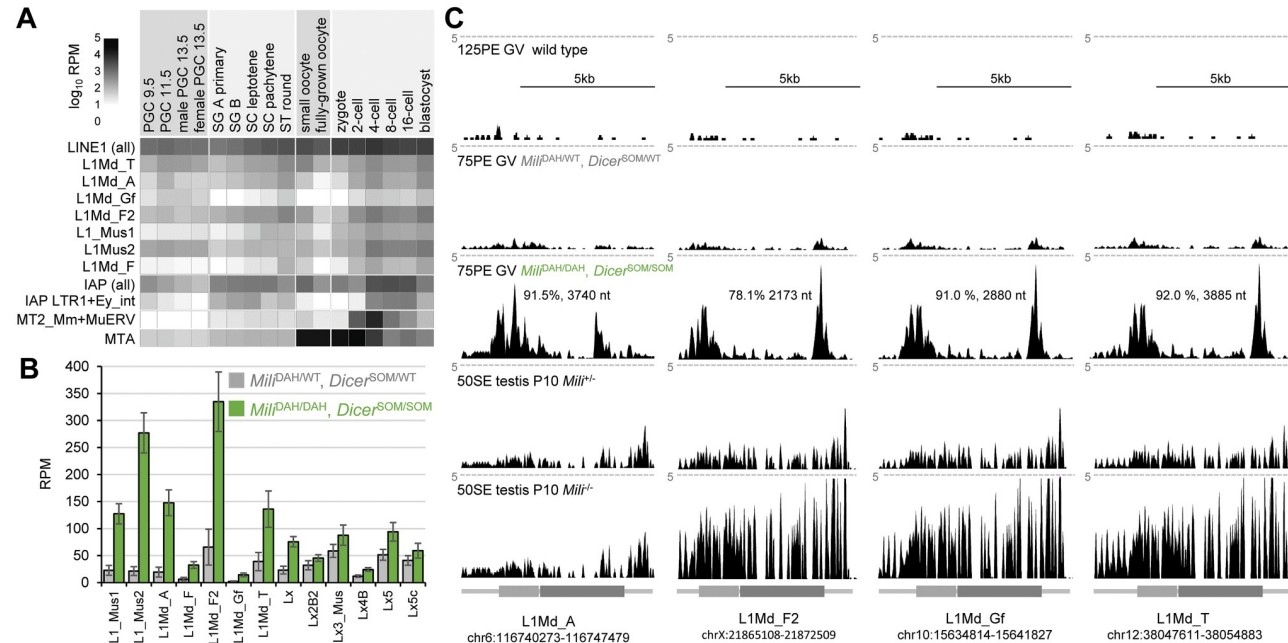

**Fig 6. Retrotransposon expression in the female germline.** (A) Relative retrotransposon expression during the germline cycle. The heatmap was produced as described previously [46] using published expression data [45, 48–50]. Depicted L1 and IAP retrotransposon subfamilies have full-length intact retrotransposon insertions in C57Bl/6 genome. (B) Selected L1 retrotransposon contribution to transcriptomes of double mutants ($Mili^{DAH/DAH}$, $Dicer^{SOM/SOM}$) and controls ($Mili^{DAH/WT}$, $Dicer^{SOM/WT}$). Four biological replicates. Error bar = SD. (C) Most expressed intact L1 retrotransposons from the four most abundant subfamilies. Shown are UCSC genome browser snapshots of specific L1 insertions with the highest coverage by RNA-seq reads in double mutants. The samples (from the top) depict merged data from 125 nucleotide paired-end (125PE) sequencing of normal full-grown GV oocytes from Horvat et al. [54], 75 nucleotide paired-end (75PE) sequencing of controls ($Mili^{DAH/WT}$, $Dicer^{SOM/WT}$) and double mutants ($Mili^{DAH/DAH}$, $Dicer^{SOM/SOM}$) from this study, and 50 nucleotide single-end (50SE) sequencing of total transcriptome of 10 days old wild-type and $Mili^{-/-}$ testes [28]. All RNA-seq data were scaled and are shown at the same scale 5 CPM indicated by the dashed line. L1 schemes are depicted with ORF1 and ORF2 as lighter and darker grey rectangle, respectively.

order of magnitude higher L1 RNA abundance (Fig 6B), these sequences could originate from transcribed insertions that cannot retrotranspose.

We thus examined expression of all full-length L1 elements with intact ORF in the reference C57Bl/6 genome in fully-grown oocytes by mapping our RNA-seq libraries directly on these elements (using RNA-seq of mutants from this work and from [54]). While RNA-seq analysis cannot directly distinguish between reads originating from "dead" elements and intact ones, coverage by perfectly mapping reads still allows to assess how strong is the evidence supporting expression of intact, retrotransposition-competent elements. We thus focused on those intact elements exhibiting not only high expression in the number of reads per element but also the highest coverage of the entire element by RNA-seq data from double mutants (depth ~ $175 \times 10^6$ perfectly matched 75 nt paired-end (75PE) sequenced fragments). We also included previous RNA-seq data from control and *Mili*-deficient postnatal day 10 (P10) testes [28].

The L1 elements showing the highest coverage by RNA-seq data in double mutants exhibited minimal coverage in normal fully grown oocytes (Fig 6C). In fact, analysis using perfectly-mapping reads from 125 nucleotide paired-end sequencing of fully-grown wild-type GV oocyte (125PE, top row in Fig 6C) revealed minimal if any expression of intact L1 elements with signal exisiting in the 5'UTR, which could be associated with L1's antisense promoter. Since ~$46.8 \times 10^6$ fragments (i.e. ~ $94 \times 10^6$ reads) in 125PE libraries were mapable with perfect complementarity on the C57Bl/6 genome, these data suggest that most retrotransposition-competent L1 elements are transcriptionally silent during oocyte growth. A similar result

was obtained with 75 PE data from heterozygous controls (depth ~144x10$^6$ fragments, Fig 6C). In contrast, coverage of intact L1 elements in *Mili* heterozygotes in RNA-seq analysis of whole testes appeared much better than in oocytes suggesting higher L1 expression during spermatogenesis (Fig 6C, bottom two sets). At the same time, the loss of *Mili* yielded a mild increase L1 RNA abundance (Fig 6C) but this could reflect the fact that *Mili*-mediated suppression of L1 in testes concerns only a subpopulation of cells and this increase may be diluted in the whole testes transcriptome.

Transcriptomes of double mutant oocytes showed increased abundance of reads mapping to specific L1 elements. However, coverage of such elements with perfectly mapping reads was never complete (92% in the best case, Fig 6C) while the longest L1 region covered by sequencing data was 3885 nt (Fig 6C). In contrast, when analyzing long L1 insertions regardless of the intact ORFs or 5' end, we could identify several elements with full or almost full coverage and even with RNA reads extending into their genomic flanks suggesting that these are *bona fide* sources of L1 transcripts (S5 Fig).

Altogether, these data imply that the bulk of reads mapping to intact full-length elements are multimapping reads originating from other loci and that full-length intact L1 elements have minimal if any expression in fully-grown oocytes even in the absence of RNAi and piRNA pathways. Consistent with this notion is that suppression of RNAi and piRNA pathways in oocytes does not lead to production of detectable L1 ORF1 protein (S6 Fig). This implies that, unlike derepression of L1 in *Mili*$^{-/-}$ testes [24], increased abundance of L1 transcripts in fully-grown double mutant GV oocytes is unlikely to represent significant L1 mobilization.

## Discussion

We experimentally tested whether or not RNAi acts redundantly and compensates the loss of piRNAs in the murine female germline. Existence of such a compensation was proposed previously [31, 32] and seemed plausible given normal fertility of female mutants of the piRNA pathway [17–19, 30] and simultaneous presence of RNAi and piRNA pathways in the oocyte [32, 40]. Furthermore, the vertebrate female germline is not insensitive to the loss of the piRNA pathway *per se* as shown in zebrafish [55]. To test the compensation hypothesis, we engineered a novel mouse mutant *Dicer*$^{SOM}$, which could be directly crossed with a piRNA pathway mutant to obtain females deficient in both pathways in the germline in a simple crossing scheme.

*Dicer*$^{SOM}$ mice express a tagged Dicer protein and lack expression of the short Dicer$^O$ variants, which support RNAi in oocytes [46]. The *Dicer*$^{SOM}$ model is superior to our earlier *Dicer*$^{AMT}$ RNAi-deficient mouse model that lacked the main *Dicer*$^O$ promoter and phenocopied oocyte-specific *Dicer* knock-out phenotype [44]. *Dicer*$^{AMT}$ model was problematic because of varying penetrance of the phenotype due to an alternative LTR promoter that could also drive Dicer$^O$ expression [46]. *Dicer*$^{SOM}$ model faithfully phenocopies the loss of Dicer in mouse oocytes including derepression of RNAi-targeted genes and retrotransposons. Low levels of the full-length Dicer remain expressed in *Dicer*$^{SOM/SOM}$ oocytes (Fig 2F). The residual full-length Dicer seems to sustain miRNA biogenesis [44] but not efficient RNAi, as shown by transcriptome remodeling and phenocopy of Dicer null phenotype. This is also consistent with minimal siRNA levels produced by the full length Dicer in somatic cells from dsRNA substrates, which are capable of inducing RNAi in oocytes [56, 57].

For a piRNA pathway mutant, we have chosen *Mili*$^{DAH}$ mouse model expressing catalytically dead MILI [24]. MILI is involved in processing of primary piRNA transcripts [58]; its prenatal or postnatal loss derails the piRNA pathway and causes male but not female sterility

[24, 59]. When we crossed *Dicer*^SOM mice with *Mili*^DAH mice, the double mutants revealed no effect on ovarian oogenesis, up to the diplotene stage of the first meiotic division, in which oocytes remain arrested until resumption of meiosis (reviewed in [60]). This shows that piRNA and RNAi pathways in the mouse female germline do not suppress an acute threat, which would manifest as an ovarian phenotype when both pathways are removed. However, long term consequences could be still sufficient for maintaining both pathways present during evolution.

Why there would be a piRNA pathway in mouse oocytes at all? As suggested during the peer review, it is possible that the piRNA pathway could be an "insurance policy" against a future retrotransposon invasion. Occasional retrotransposon invasions occurring during rodent evolution could represent sufficient selective pressure for maintaining maternal piRNA pathway. Furthermore, common germline transcription factors, such as SOHL1/2 could contribute to the presence of the piRNA pathway in male and female germlines regardless how significant would be its presence in the female germline. This notion would be supported by reduced levels of many transcripts encoding piRNA pathway components in SOHLH and SOHLH2 female mutants [61] (S7 Fig).

Importantly, we observed that RNAi and piRNA pathways suppress retrotransposons in oocytes in a partially redundant manner, which could be explained by the stochastic nature of evolution of substrates giving rise to siRNAs and piRNAs. The piRNA pathway is an adaptive defense, where the recognition of an invading retrotransposon is stochastic; it requires retrotransposon insertion into one of the piRNA clusters yielding piRNA production and retrotransposon repression [62]. The post-transcriptionally acting RNAi also evolves stochastically in a different way. RNAi requires dsRNA, which can form when retrotransposon insertions form a transcribed inverted repeat or are transcribed antisense to produce RNAs basepairing with retrotransposon transcripts. Furthermore, some retrotransposons have intrinsic potential to produce dsRNA. In particular, L1 carries an antisense promoter in its 5'UTR and can produce sense and antisense RNAs with potential to basepair and trigger RNAi (reviewed in [63]). Notably, the antisense transcription from L1 5'UTR, which exists across mammals, may represent L1's own strategy for maintaining minimal expression levels sufficient for L1 retrotransposition in *cis* [11]; natural selection would otherwise yield L1s lacking the antisense promoter if it would significantly reduce L1's fitness.

It could be that L1 is not active during oocyte development because transcription factors controlling its transcription are present during different parts of the germline cycle. However, this would be inconsistent with the proposed role of L1 during fetal oocyte attrition, which requires variable L1 expression [64]. Thus, it is more plausible that L1 is transcriptionally repressed in the female germline by some piRNA-independent mechanism. One of the candidate transcriptional silencing mechanisms, which can evolve and act in parallel to the piRNA pathway are Krüppel-associated box domain zinc finger proteins (KRAB-ZFPs, reviewed in [65]), whose C-terminal arrays of DNA binding zinc fingers have potential to evolve into specific retrotransposon repressors. Their cofactor, TRIM28 was found enriched on some L1 subfamilies, particularly L1MdF and L1MdF2 [66]. Furthermore, KRAB-ZFP *Gm6871* was proposed to selectively target L1MdF2 elements in mice [66]. *Trim28* and *Gm6871* are expressed in 5dpp oocytes [45] making the KRAB-ZFP system an excellent candidate for a parallel functional protection of the mouse female germline, which prevents mobilization of L1 elements in the absence of RNAi and piRNA pathways.

It should be highlighted that any interpretation of the piRNA pathway function and its loss-of-function phenotype must take into the account that the threat represented by mobile elements varies along the germline cycle and changes during evolution in species-specific manner. In any case, while there is no conclusive answer to the question why is the female

germline insensitive to the loss of the piRNA pathway, our results rule out RNAi as a responsible mechanism and indicate that additional transcriptional silencing is involved in retrotransposon repression during oocyte development.

## Methods

Numerical data for graphs are provided in the S1 File.

### Ethics statement

Animal experiments were approved by the Institutional Animal Use and Care Committee (approval no. 34–2014) and were carried out in accordance with the law.

### Animals

*Mili*[DAH] mice were kindly provided by Donal O'Carroll; this strain was characterized previously [24]. *Dicer*[SOM] mice were produced in the Transgenic Unit of the Institute of Molecular Genetics ASCR, Czech Centre for Phenogenomics.

For genotyping, tail biopsies were lysed in PCR friendly lysis buffer with 0.6 U/sample Proteinase K (Thermo Scientific) at 55 ˚C, with 900 rpm shaking until dissolved (approx. 2.5 h). Samples were heat-inactivated at 75 ˚C, 15 min. The lysate was diluted three times and 1 μl used with HighQu DNA polymerase (0.5 U/reaction) master mix for PCR. Genotyping primers are provided in S2 Table. PCR program was as follows– 95 ˚C, 3 min, 38 x [95 ˚C, 30 s, 58 ˚C, 30 s, 72 ˚C, 1 min], 72 ˚C, 3 min. PCR products were analyzed using a 1.5% agarose gel electrophoresis.

### Generation of Dicer[SOM] mice

We first produced ESCs with the *Dicer*[SOM] allele and then used those for producing chimeric mice and establishing *Dicer*[SOM] line upon germline transmission of the *Dicer*[SOM] allele. *Dicer*[SOM] allele in ESCs [67] was generated by using CRISPR-Cas9 [68] mediated modification of the endogenous *Dicer* locus. Pairs of sgRNAs were designed to cleave *Dicer* genomic sequence in intron 2 (sequence of DNA targets: mDcr_i2a 5'-GTACCCAAATGGATAGAA-3', mDcr_i2b 5'-GTTGGGATGGAGGTTGTT-3') and intron 6 (sequence of DNA targets: mDcr_i6a 5'-ACTACGCTAGGTGTAAACAG-3', mDcr_i6b 5'-TGCAGTCCCCGGACGTT AAAT-3'). A template for homologous recombination was designed to contain an HA-tag at the N-terminus of *Dicer* coding sequence and DNA sequence of exon 2 to exon 7 of *Dicer* (Fig 2A). Detailed description of preparation of the template for homologous recombination and its sequence are provided in the supplement (S1 Text).

Upon screening ~600 ESC clones by PCR genotyping and Western blotting, we selected four ESC lines (S1 Fig), which were used to produce chimeric mice. *Dicer*[SOM] ESCs were injected into eight-cell stage of C57Bl/6 host embryos. In total, 255 chimeric embryos were constructed, which were transferred into pseudopregnant foster mothers of ICR strain background. Of 27 weaned animals, fur color indicated chimerism in seven males and one female. Five males showing the strongest chimerism were mated to C57Bl/6 females to obtain germline transmission. From ~ 140 genotyped progeny, we obtained one positive animal derived from the ESC clone 11 (S1 Fig), which was subsequently crossed on the C57Bl/6 background to establish the *Dicer*[SOM] line.

## Oocyte collection

Fully-grown GV oocytes were obtained from superovulated or non-stimulated 12–16 weeks old C57Bl/6 mice as described previously [69]. Resumption of meiosis was prevented with 0.2 mM 3-isobutyl-1-methyl-xanthine (IBMX, Sigma). MII oocytes were obtained from animals superovulated with 7 IU of PMSG administered on day 1 and 7 IU of hCG administered 46–48 h post PMSG, mice were sacrificed 16 h post hCG. Oocytes were collected from oviducts, cumulus cells were removed by incubation with 0.1 uM hyaluronidase solution in M2 media for 3–5 min, oocytes were washed from hyaluronidase in M2 media and collected in PBS-PVP.

## Cell culture and transfection

Mouse R1 ESCs were cultured in LIF media: KO-DMEM supplemented with 15% fetal calf serum, 1x L-Glutamine (Invitrogen), 1x non-essential amino acids (Invitrogen), 50 μM β-Mercaptoethanol (Gibco), 1000 U/mL LIF (ISOKine, ORF Genetics), penicillin (100 U/mL), and streptomycin (100 μg/mL). For transfection, cells were plated on a 24-well plate, grown to 50% density and transfected using Lipofectamine 3000 (Thermo Fisher Scientific) according to the manufacturer's protocol.

## Western blotting

Mouse tissues were homogenized mechanically in RIPA lysis buffer (20 mM HEPES (pH 7.8), 100 mM NaCl, 1 mM EDTA (pH 8.0), 0.5% IGEPAL-25%, 1 mM fresh DTT, 0.5 mM PMSF, 1 mM NaF, 0.2 mM $Na_3VO_4$, supplemented with 2x protease inhibitor cocktail set (Millipore)), centrifuged for 15 min, at 16 000 g, 4 ˚C and supernatant was used for protein electrophoresis. Protein concentration was measured by Bradford assay, 40 μg of protein was loaded per late. ESCs were grown in 6-well plates. Before collection, cells were washed with PBS and lysed in RIPA lysis buffer with inhibitors.

Proteins were separated on 5.5% polyacrylamide gel and transferred on PVDF membrane (Millipore) using semi-dry blotting for 50 min, 35 V. The membrane was blocked in 5% skim milk in TTBS, Dicer was detected using rat anti-HA 3F10 monoclonal primary antibody (Sigma-Aldrich) diluted 1:2 500, membrane was incubated overnight at 4 ˚C in blocking solution. Washed in TTBS buffer, incubated with secondary antibody (HRP-anti-Rat diluted 1:50 000 in TTBS). Incubated for 1h at room temperature. Washed in TTBS, signal was detected using Supersignal west femto substrate (Thermo Scientific). For tubulin detection, samples were run on 10% PAA gel and incubated with anti-Tubulin (Sigma, #T6074) mouse primary antibody diluted 1:5 000 and anti-mouse-HRP secondary antibody 1:50 000.

## qPCR analyses

Ten oocytes per one reverse transcription (RT) reaction were collected in 1 ul PBS-PVP. 20 U of Ribolock RI (Thermofisher), 1 ug of yeast total RNA (carrier RNA) and RNase free water up to 5 ul was added and oocytes were incubated at 85 ˚C for 5 min for lysis. Crude lysate was used for RT with Maxima H Minus Reverse Transcriptase (Thermo Scientific). Ten oocytes from the same mouse were used for a control RT minus reaction. cDNA equivalent of a half of oocyte was used per a qPCR reaction. Maxima SYBR Green qPCR Master Mix (Thermo Scientific) was used for qPCR. qPCR was performed in technical triplicates for each biological replicate. For expression levels, we first estimated average values of the technical replicates relative to *Hprt* levels for each biological replicate and then we calculated average and standard deviation values for biological replicates.

## Histology

Ovaries were fixed in Bouin solution for 16–20 h. Next, they were washed 3x 5 min with PBS, 30 min in 50% EtOH and transferred in 70% EtOH (all steps at 4˚C). Whole ovaries were dehydrated, cleared and perfused with molten paraffin during histological processing. Ovaries were then embedded in paraffin blocks and sectioned to 8 μm sections and stained with conventional hematoxylin and eosin stain. Testes were fixed in Davidson's solution for 16–20 h and processed the same way as ovaries. 6 μm tissue sections were used for Dicer detection.

## Immunofluorescent staining

For analyzing microtubule defects, oocytes were isolated 16 h post hCG, fixed for 1h at room temperature in 4% PFA, permeabilized in 0.1% Triton X-100 for 10 min at room temperature (RT), washed in 2% BSA, 0.01% Tween-20 in PBS for 3x10 min, Blocked in 2% BSA, 0.01% Tween-20 in PBS for 1h, stained by primary antibody anti-β-Tubulin-Cy3 (TUB 2.1) (Abcam, ab11309) 1:200 for 1 h at room tempertaure, washed 3x 10 min in 0.01% Tween-20 in PBS, stained by 1 μg/ml DAPI for 5 min, mounted in Vectashield (Vector laboratories). Images of oocytes were acquired on DM6000 upright wide field microscope, images were processed by ImageJ. The experiment was performed twice, each time with four 17–19 weeks old *Dicer*^SOM/SOM females.

For detection of HA-tagged Dicer, slides with 6 μm testis sections were deparaffinized and subjected to antigen retrieval in citrate buffer (pH = 6). Sections were then permeabilized with 0.1% Triton X-100 for 15 min, blocked in 10% normal donkey serum, 0.1M glycine, 2% BSA for 1h at room tempertaure, incubated with primary antibody in blocking buffer (polyclonal rabbit anti-HA antibody (Cell signaling, 1:100) overnight at 4 ˚C, incubation with secondary antibody in blocking buffer for 1 h at room temperature in the dark (donkey anti-rabbit antibody conjugated with Alexa 488) diluted 1:800. Nuclei were stained with 1 μg/ml DAPI for 5 min, slides were mounted in Vectashield media (Vector labs).

## RNA sequencing

Total RNA was extracted from 25 GV oocytes using Arcturus Picopure RNA isolation kit according to the manufacturer's protocol. RNA was eluted in 11 μl of RNAse free water. Integrity of RNA was confirmed using 2100 Bioanalyzer (Agilent Technologies). RNA-seq libraries were generated using V2 version of Ovation RNA-seq kit from Nugen. Libraries were prepared according to the manufacturer's protocol. cDNA fragmentation was performed on Bioruptor sonication device (Diagenode) as follows: 45 s ON, 30 s OFF for 21 cycles on low intensity, 100 ng of fragmented cDNA was used for library preparation. Libraries were amplified by 9 cycles of PCR and quantified by Qubit HS DNA kit, their quality was assessed using 2100 Bioanalyzer HS DNA chip. RNA-seq libraries were pooled and sequenced by 50 nt single-end reading using the Illumina HiSeq2000 platform at the Genomics Core Facility at EMBL. The RNA sequencing which included double mutants was sequenced by 75 paired-end sequencing using the Illumina NextSeq500 platform at the genomics core at IMG RNA-seq data were deposited in the Gene Expression Omnibus database under accession ID GSE132121.

## Bioinformatic analyses

**RNA-seq mapping and expression analysis.** All RNA-seq data were mapped onto indexed genome using STAR 2.5.3a [70]: *STAR—readFilesIn ${FILE}.fastq.gz—genomeDir $REF_GENOME_INDEX—runThreadN 8—genomeLoad LoadAndRemove—limitBAMsortRAM 20000000000—readFilesCommand unpigz—c—outFileNamePrefix ${FILE}.—*

*outSAMtype BAM SortedByCoordinate—outReadsUnmapped Fastx—outFilterMultimapNmax 99999—outFilterMismatchNoverLmax 0.2—sjdbScore 2.*

Read files were mapped onto mouse genome version mm10/GRCm38. Mapped reads were counted using program featureCounts [71]: *featureCounts -a GENCODE.M15 -F ${FILE}—minOverlap 10—fracOverlap 0.00 -s 0 -M -O—fraction -J -T 8 ${FILE}.bam.*

The GENCODE gene set (GENCODE M15) was used for the annotation. The count values were normalized by *rlog* function from program suite DESeq2 [72], which normalizes the data to the library size. The values of log2-fold-change were calculated among all the replicates of the corresponding conditions. The median was reported as the final value. Data were visualized as scatterplots.

**Correlation matrices.** Correlation matrices were calculated on the same count values normalized to per kilobase per million (RPKM) units. The retrotransposons were annotated by RepeatMasker, the mapped reads were counted for the GROUP category defined in *RepeatMasker-retrotransposon-groups.csv* (S3 Table). The values were normalized to RPKM.

Small RNA-seq data were mapped onto indexed genome using STAR 2.5.3a: *STAR—readFilesIn ${FILE}.fastq.gz—genomeDir $REF_GENOME_INDEX—runThreadN 8—genomeLoad LoadAndRemove—limitBAMsortRAM 20000000000—readFilesCommand unpigz -c—outFileNamePrefix ${FILE}.—outSAMtype BAM SortedByCoordinate—outReadsUnmapped Fastx—outFilterMismatchNmax 1—outFilterMismatchNoverLmax 1—outFilterMismatchNoverReadLmax 1—outFilterMatchNmin 16—outFilterMatchNminOverLread 0—outFilterScoreMinOverLread 0—outFilterMultimapNmax 9999—alignIntronMax 1—alignSJDBoverhangMin 999999999999*

The histograms of the read lengths were calculated on the primary read alignments for the GROUP category defined in *RepeatMasker-retrotransposon-groups.csv* (S3 Table). The potential siRNA targets (S4 Table) were manually curated using the small RNA-seq datasets [32, 41, 43] and selected annotated protein-coding genes for which (i) existed antisense endogenous 21-23nt small RNAs with > 50 RPM average abundance and (ii) for which we could identify their origin.

**Retrotransposon expression analysis.** For retrotransposon expression heatmap in germline (Fig 5A), expression data from four different published datasets were mapped onto genome as described above GSE35005 [48]; GSE70116 [45], GSE45719 [49], GSE41908 [50], and GSE70731 [28]. Next, reads overlapping one of the chosen retrotransposon subfamilies annotated by RepeatMasker [12] were counted with the following restrictions: (1) only perfectly mapping reads (i. e. nM tag in SAM files = 0) were counted, (2) only reads completely overlapping retrotransposon coordinates were counted, (3) coordinates of retrotransposons overlapping with genes annotation from Ensembl database (version 93) were removed from counting, and (4) each sequencing read mapped to more than one genomic sequence belonging to the same retrotransposon subfamily was counted only once. Count values were then normalized for library size using number of perfectly mapping reads in millions to obtain RPM values. For experiments with multiple replicates the RPM was calculated as the mean RPM value over all replicates.

In order to examine expression of 2686 full-length or nearly full-length L1 elements with ORF1 and ORF2 coding sequences of expected lengths (Fig 6C; S5 Table in fully-grown GV oocytes (125 PE from GSE116771 [54] and 75 PE from this work), RNA-seq reads were mapped onto genome using BBMap 38.63 (http://bbmap.sourceforge.net/) with only read pairs matching reference perfectly with no substitutions or indels allowed. All multimapping reads were retained for the analysis:

*bbmap.sh -Xmx60g threads = 24 in = ${FILE}_1.fastq.gz in2 = ${FILE}_2.fastq.gz outm = ${FILE}.sam outu = ${FILE}.unmapped.fq.gz perfectmode = t ambiguous = all pairedonly = t maxsites = 20000 nhtag = t amtag = t scoretag = t unpigz = t.*

## Supporting information

**S1 Text. Supplementary methods and list of supporting information.**
(DOCX)

**S1 Fig. Analysis of ESC lines for *Dicer*^SOM^ mouse model production by PCR and Western blotting.** The yellow marked ESC lines were used for producing chimeric mice, the clone 11 (G2) gave rise to *Dicer*^SOM^ animals used in the experiment.
(TIFF)

**S2 Fig. Additional micrograps documenting spindle defects in meiotically maturing Dicer-SOM/SOM oocytes.** Framed micrographs were used for Fig 2A. Size bar = 10 μm.
(TIFF)

**S3 Fig. Comparison of transcriptomes of oocytes affected by different mutations in RNAi pathway.** (A) Pairwise comparisons of relative gene expression changes using DESeq2-normalized expression values. In red are depicted genes potentially targeted by RNAi (i.e. with identified antisense siRNAs). (B) Correlation matrices calculated from RPKM values all annotated genes (upper panel) and predicted siRNA targets (lower panel).
(TIFF)

**S4 Fig. Comparison of transcriptomes of oocytes affected by mutations of RNAi and/or piRNA pathway.** MA plots with the same Y-scale depict changes relative to controls (*Mili*^DAH/WT^, *Dicer*^SOM/WT^) and double mutant (*Mili*^DAH/DAH^, *Dicer*^SOM/SOM^) relative to RNAi mutant (*Mili*^DAH/WT^ *Dicer*^SOM/SOM^). Transcripts with significantly higher and lower transcript levels are shown in red and blue, respectively. Transcripts from a tandemly duplicated cluster at chromosome 12 are depicted as triangles.
(TIFF)

**S5 Fig. Selected long but not necessarily intact L1 retrotransposons from selected abundant subfamilies with the best coverage in RNA-seq data from double mutant data.** Shown are UCSC genome browser snapshots of specific L1 insertions, their % coverage and the longest covered sequence by RNA-seq data in double mutants. The samples (from the top) depict merged data from 125 nucleotide paired-end (125PE) sequencing of normal full-grown GV oocytes from Horvat et al. [54], 75 nucleotide paired-end (75PE) sequencing of controls (*Mili*^DAH/WT^, *Dicer*^SOM/WT^) and double mutants (*Mili*^DAH/DAH^, *Dicer*^SOM/SOM^) from this study, and 50 nucleotide single-end (50SE) sequencing of total transcriptome of 10 days old wild-type and *Mili*^--/--^ testes[28]. All RNA-seq data were scaled and are shown at the same scale 5 CPM indicated by the dashed line. The scheme below L1Md_A depicts intact ORF1 and ORF2 as lighter and darker grey rectangle, respectively.
(TIFF)

**S6 Fig. Lack of positive staining for L1 ORF1 in fully-grown oocytes.** Oocytes were stained with α-L1-ORF1 antibody ([73] generous gift from Donal O' Carroll) and analyzed by confocal microscopy. Shown are 10 μm optical sections. L1 ORF1 signal is in red channel, DNA stained with DAPI is shown in blue color. Size bar = 30 μm.
(TIFF)

**S7 Fig. Expression of piRNA pathway factors in mouse newborn ovaries lacking *Sohlh1* and *Sohlh2*.** RNA-seq data were obtained from a published dataset PRJNA293873 [61] and gene expression was analyzed as described in Methods.
(TIFF)

**S1 Table. Differentially expressed genes in RNAi & piRNA mutant/RNAi mutant.**
(XLSX)

**S2 Table. Primer table.**
(XLSX)

**S3 Table. RepeatMasker retrotransposon groups.**
(CSV)

**S4 Table. siRNA target list.**
(CSV)

**S5 Table. Full length L1 table.**
(CSV)

**S1 File. Numerical data for graphs.**
(XLSX)

## Acknowledgments

We thank Vladimir Benes and EMBL sequencing facility and Michal Kolar and IMG Functional Genomics and Bioinformatics Core for help with RNA-seq experiments, Dagmar Zudova from the Czech Center for Phenogenomics for help with histology analysis, Kristian Vlahovicek (Zagreb University, Croatia) for providing hardware support for bioinformatics analysis, Donal O'Carroll (MRC Centre for Regenerative Medicine, University of Edinburgh, UK) for providing *Mili*[DAH] mice and α-L1-ORF1 antibody. Support of E.T. and F.H. was in part provided by the Charles University through a PhD student fellowship; this work will be in part used to fulfill requirements for a PhD degree and hence can be considered "school work".

## Author Contributions

**Conceptualization:** Eliska Taborska, Radek Malik, Petr Svoboda.

**Data curation:** Josef Pasulka, Filip Horvat.

**Formal analysis:** Josef Pasulka, Filip Horvat, Zoe Jelić Matošević.

**Funding acquisition:** Petr Svoboda.

**Investigation:** Eliska Taborska, Radek Malik.

**Methodology:** Eliska Taborska, Irena Jenickova.

**Project administration:** Eliska Taborska, Radek Malik, Petr Svoboda.

**Software:** Josef Pasulka, Filip Horvat, Zoe Jelić Matošević.

**Supervision:** Radek Malik, Petr Svoboda.

**Validation:** Josef Pasulka.

**Visualization:** Eliska Taborska, Josef Pasulka, Filip Horvat.

**Writing – original draft:** Petr Svoboda.

**Writing – review & editing:** Eliska Taborska, Josef Pasulka, Radek Malik, Filip Horvat, Petr Svoboda.

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
