## [Decision Letter · Decision Letter 0]

7 Aug 2019

Dear Dr Svoboda,

Thank you very much for submitting your Research Article entitled 'Restricted and non-essential redundancy of RNAi and piRNA pathways in mouse oocytes' to PLOS Genetics. Your manuscript was fully evaluated at the editorial level and by independent peer reviewers. The reviewers appreciated the attention to an important problem, but raised some substantial concerns about the current manuscript. Based on the reviews, we will not be able to accept this version of the manuscript, but we would welcome a much-revised version of your paper if these issues can be addressed satisfactorily. 

If you decide to revise the manuscript for further consideration at PLOS Genetics, please aim to resubmit within the next 60 days, unless it will take extra time to address the concerns of the reviewers, in which case we would appreciate an expected resubmission date by email to plosgenetics@plos.org.

[LINK]

We are sorry that we cannot be more positive about your manuscript at this stage. Please do not hesitate to contact us if you have any concerns or questions.

Yours sincerely,

Lin He

Associate Editor

PLOS Genetics

Gregory Barsh

Editor-in-Chief

PLOS Genetics

Reviewer's Responses to Questions

**Comments to the Authors:**

Reviewer #1: Taborska et al. examined redundant roles of two small RNA pathways, RNAi and piRNAs, in the silencing of retrotransposons in mouse oocytes. Specifically, this works asked if RNAi activity can explain the lack of apparent germ cell phenotypes in piRNA-deficient mouse oocytes. To address this question, the authors:

1. generated an RNAi pathway mutant mice lacking Dicer protein (Dicer SOM). This is a tighter Dicer mutant allele compared to the one created in their previous study on the oocyte-specific Dicer isoform;

2. characterized phenotypes of the Dicer SOM mice;

3. generated double-mutant Dicer SOM; Mili and control animals;

4. evaluated retrotransposon expression in single and double mutant combinations.

The main finding of this work is that lack of prominent female germ cell phenotypes in the absence of piRNAs cannot be explained by the redundancy with the RNAi pathway. A minor observation is that retrotransposon silencing by the individual small RNA pathways do not overlap completely. As such, this work clarifies the individual and combined contributions of the two small RNA pathways to retrotransposon silencing in the female germline. However, the study represents an incremental advancement in our understanding of retrotransposon regulation in the female germline. Crucially, the primary mechanism(s) that ensures retrotransposon silencing in mouse oocytes remains unknown.

Major comments:

1. Small RNAseq data in figure one is in from Yang et al., 2016. Small RNAseq data should be acquired for single and double mutants to validate putative RNAi targets from mRNA seq, as well as show changes in global production of siRNA and piRNA with mutations.

2. Aside from eliminating truncated Dicer variant, the Dicer SOM mouse doesn't seem to significantly change the overall phenotype compared to Dicer deltaMT, except from few putative RNAi targets and upregulated genes in RNA-seq. Not really significant or novel that this mouse was generated.

3. What is a "genomic checkpoint" that the piRNA pathway relies on? Describe better as this is mentioned twice.

4. Explain better the significance of including comparisons to Ago2, and differences in Dicer vs Ago2 oocyte phenotypes?

Minor comments:

Figure 2B. Missing a loading control.

Figure 2C. Needs labels, what is green and blue? What is the genotype?

Figure 2E. Endogenous control gene for qPCR?

Figure 3B. n oocytes are from how many females?

Figure 3C. Reference for initial set of genes that are sensitive to RNAi pathway?

Figure 3D. Need to validate putative RNAi targets using small RNAseq. Otherwise gene expression differences could be indirect and due to oocyte phenotypes.

Figure 3E. What is the difference between Ago2 mutants from Stein et al., 2015 in orange vs. pink? Triangle vs. square symbols?

Figure 4A. More detailed quantification of primordial follicles vs advanced follicles, maybe there is a difference at that resolution between WT and mutants.

Figure 4D. Better to use RNAseq to analyze abundance of TEs in mutants to eliminate background elements within genes that are changing as a consequence of mutation. Also, how many biological replicates per bar in qRT-PCR?

Reviewer #2: Mammalian PIWI-piRNA pathways have been studied mostly in mouse. Mice express three PIWIs (MIWI/PIWIL1, MIWI2/PIWIL4, and MILI/PIWIL2). All three PIWI proteins are expressed at different stages during spermatogenesis, but only MILI is expressed, albeit weakly, in female germ cells. Deficiency in MIWI, MILI or MIWI2 leads to the activation of transposable elements (TEs) including L1 and IAP elements, and spermatogenesis is arrested at specific stages of spermatogenesis. Mutations in mouse PIWI genes affect the male germline but not the female germline. The authors previously identified an oocyte-specific truncated Dicer isoform, termed Dicero. This Dicer isoform produces endogenous siRNAs and together with AGO2 targets TEs and regulates their activity. These findings have led to assumptions that PIWI-piRNA pathways do not play a role during mouse oogenesis. To address the significance of redundancy of endogenous siRNA and piRNA pathways in oocytes, the authors have generated a mouse model lacking Dicero and then crossed it with MiliDAH mice expressing catalytically inactive MILI. In this study, the authors find that the simultaneous loss of endogenous siRNA and piRNA pathways does not affect oocyte development. The authors also find that endo-siRNA and piRNA pathways suppress some but not all TEs in a redundant manner. This raises the possibility that yet another TE silencing mechanism must operate in female germ cells to maintain genome integrity. Although their findings are themselves not surprising and completely expected, studies of this sort are good references and resources for further comparisons.

Minor comments:

1. The quality of images used in Figure 2C is poor. Its legend lacks important information such as which panel shows what with which antibody.

2. The quality of images used in Figure 3A is also poor.

3. Figure 3C: The authors should add non-targeted genes in this experiment.

4. Figure 4A – C: here the authors analyze ovarian weight, histology and number of fully-grown GV oocytes using mice with age between 8 and 18 weeks. The authors should describe in legends the age in weeks of mice they use for each experiment. In my opinion, the authors should use mice with same age in weeks in these experiments. In addition, the authors’ better present data with male mice in Figure 4D for the reader to understand the degree of contribution of RNAi and piRNA pathways to suppress TEs in female gonads.

Reviewer #3: The manuscript by Taborska et al genetically explores the combined roles of the RNAi and piRNA pathways in the female mouse germ cell development through the use of advanced mouse genetics. The study is designed to address the question of whether the RNAi pathway masks a function for the piRNA pathway in the female mouse germline. An elegant new allele of Dicer is generated that forces the expression of somatic Dicer isoform throughout the entire animal inclusive of the germline. These Dicer-som are viable but present female-specific infertility. The combination of Dicer-som with the catalytically inactive MILI results in the same developmental phenotype but has a distinct molecular phenotype comparted to Dicer-som mice. This stringent genetic experiment and the molecular analysis undertaken substantiate the principal claims of the manuscript. I am intrigued by the data of Figure 5. Overall, I am supportive of publication but have concerns to be addressed:

The upregulation of RNAi targets in figure 4D & F could also be possibly presented in cumulative frequency plots (S-curves). These have traditionally been used for miRNA effects and are quantitative and very easy to understand.

The data presented in Figure 5 is not presented in the results section. This needs to be rectified.

The data presented in Figure 5 is an extremely important observation. I think it should be given some mention in the abstract. I also think the key message from the low levels of L1 transcription in the female germline and minor upregulation of L1 transcripts (compared to the male germline) in the absence of the RNAi and/or piRNA pathways would be substantiated by staining for L1 ORF1 protein. The expectation would be that it should not be detected in the respective mutant genotypes.

I think the discussion should give a sentence or two on why the piRNA-pathway expression in should be retained in the female germline of species that does abundantly express active TEs therein. The piRNA pathway is more than just the PIWI protein but requires the expression of many biogenesis factors (at least >10). Could the expression of the pathway be an insurance policy against a future invasion?

**Have all data underlying the figures and results presented in the manuscript been provided?**

Reviewer #1: Yes

Reviewer #2: Yes

Reviewer #3: Yes

PLOS authors have the option to publish the peer review history of their article (what does this mean?). If published, this will include your full peer review and any attached files.

Reviewer #1: No

Reviewer #2: No

Reviewer #3: No

---

## [Decision Letter · Decision Letter 1]

15 Nov 2019

Dear Dr Svoboda,

Thank you very much for submitting your Research Article entitled 'Restricted and non-essential redundancy of RNAi and piRNA pathways in mouse oocytes' to PLOS Genetics. Your manuscript was fully evaluated at the editorial level and by independent peer reviewers. The reviewers appreciated the attention to an important topic but identified some aspects of the manuscript that should be improved.

The reviewers comments are largely positive at this stage, and there are only a few comments that need to be addressed. We hope to receive a revised manuscript to incorporate these suggestions, before making an editorial decision on the paper. Your revisions should address the specific points made by reviewer 1 and 3.

We hope to receive your revised manuscript within the next 7 days. If you anticipate any delay in its return, we would ask you to let us know the expected resubmission date by email to plosgenetics@plos.org.

[LINK]

Yours sincerely,

Lin He

Associate Editor

PLOS Genetics

Gregory Barsh

Editor-in-Chief

PLOS Genetics

Reviewer's Responses to Questions

**Comments to the Authors:**

Reviewer #1: Overall, I am satisfied with authors' responses.

I still have an issue with the genomic "checkpoint". The current draft describes piRNA clusters as "... specific genomic regions serving as "checkpoints" sensing invading mobile elements". This is not entirely correct since a genomic region senses nothing by itself. If not expressed, its resident sequences are invisible to the piRNA system and the cell. It is an unnecessary invention that will only confuse readers, and I ask authors to refrain from using this term. For example, the same idea can be expressed as follows:

"The piRNA pathway relies on specific genomic regions (piRNA clusters) whose expression primes piRNA production".

Reviewer #2: On the whole it has been improved and the authors have addressed some of the reviewers’ concerns. However, it is already known that the deficiency of PIWI genes results in no discernible phenotype in female mice. In contrast, the authors previously found that the deficiency of the oocyte-specific Dicer variant causes frequent meiotic spindle defects upon resumption of meiosis. Thus, it is still not clear the rationale for why the authors try to examine possible redundancy between RNAi and piRNA pathways in mouse oocytes. The authors had better focus on characterization of Dicersom/som mutant oocytes in my opinion.

Other points:

1. Figure 2 E & F show that expression levels of the full-length Dicer are markedly affected in Dicersom/som mutant oocytes. Why?

2. Figure 3A: the authors should show the figures provided to response to Reviewer #2 comments as supplementary figures.

3. The figure showing L1 ORF1 staining in the authors response to reviewer #3 appears to show that part of chromosomes (DAPI staining) is already excluded from the nucleus in Dicer mutants. This may imply that the deficiency of the Dicer activity already causes chromosome abnormality in early oocyte development and thus, the spindle defects observed are secondary.

4. The figure showing expression levels of piRNA pathway genes in oocytes contains very important information in the field. Thus, the authors should show the figure in the current manuscript. Studies of this sort are good references and resources for further comparisons.

Reviewer #3: All my concerns have been addressed.

**Have all data underlying the figures and results presented in the manuscript been provided?**

Reviewer #1: Yes

Reviewer #2: Yes

Reviewer #3: Yes

PLOS authors have the option to publish the peer review history of their article (what does this mean?). If published, this will include your full peer review and any attached files.

Reviewer #1: No

Reviewer #2: No

Reviewer #3: No

---

## [Editor Report · Decision Letter 2]

2 Dec 2019

Dear Dr Svoboda,

We are pleased to inform you that your manuscript entitled "Restricted and non-essential redundancy of RNAi and piRNA pathways in mouse oocytes" has been editorially accepted for publication in PLOS Genetics. Congratulations!

Yours sincerely,

Lin He

Associate Editor

PLOS Genetics

Gregory Barsh

Editor-in-Chief

PLOS Genetics

Comments from the reviewers (if applicable):

**Data Deposition**

http://datadryad.org/submit?journalID=pgenetics&manu=PGENETICS-D-19-00984R2

**Press Queries**

---

## [Editor Report · Acceptance letter]

13 Dec 2019

PGENETICS-D-19-00984R2 

Restricted and non-essential redundancy of RNAi and piRNA pathways in mouse oocytes 

Dear Dr Svoboda, 

We are pleased to inform you that your manuscript entitled "Restricted and non-essential redundancy of RNAi and piRNA pathways in mouse oocytes" has been formally accepted for publication in PLOS Genetics! Your manuscript is now with our production department and you will be notified of the publication date in due course.

With kind regards,

Matt Lyles

PLOS Genetics

On behalf of:
